# Analysis and Prediction of the Coupling and Coordinated Development of Green Finance–Environmental Protection in China

**Shanshan Li, Gaoweijia Wang, Li Yang \*, Jichao Geng and Junqi Zhu**

School of Economic and Management, Anhui University of Science and Technology, Huainan 232000, China
\* Correspondence: lyang@aust.edu.cn; Tel.: +86-139-5644-6008

**Abstract:** Green finance is an important tool to help China accelerate the process of environmental protection, but the level of coupling and coordinated development between it and environmental protection has not yet been explored. This study measures the coupling and coordinated level of the green finance–environmental protection system (GE system) in 30 Chinese provinces from 2011 to 2020 and uses the improved GM (1,1) model based on background value optimization to predict the future development trend of the coupling and coordinated level of the green finance–environmental protection system. The results show that: (1) the national average coupling and coordinated level of the green finance–environmental protection system has been in mild disharmony from 2011 to 2020 all the time, and only Guangdong Province and Zhejiang Province among them have reached the coordination level. (2) The coupling and coordinated level of the green finance–environmental protection system based on regional differences has a large gap, ranking in order: Eastern region > Central region > Western region > Northeast region, where the first-ranked Eastern region leads the rise, while the last-ranked Northeast region even shows a decreasing trend year by year. (3) The national average coupling and coordinated level will reach the coordination level in 2077, which fails to get ahead of the "2060 carbon neutrality" goal. Additionally, from the regional division, the Eastern region will be the first to reach the coordination level (2040), the Central region will reach the coordination level in 2043, the Western region is difficult to reach the coordination level, and the Northeast region shows the deterioration of coupling and coordinated degree, and the regional differences are still obvious. This study aims to reduce regional disparities, improve the coupling and coordinated development level of the green finance–environmental protection system nationwide, and implement the process of green development in China.

**Keywords:** green finance; environmental protection; GE system; GM (1,1) prediction model

## 1. Introduction

With the global climate warming and frequent occurrence of environmental problems, such as air pollution, water pollution, forest destruction, and so on, as well as increasingly serious environmental deterioration, strengthening ecological environmental protection and promoting global environmental governance have become major issues of universal concern for all countries around the world [1,2]. China's energy structure is dominated by coal, and its coal production has also steadily increased in recent years [3]. According to the latest statistics released by BP World Energy Statistical Yearbook, it is known that the total global coal production in 2021 is 8.173 billion tons, while China's coal production is 4.126 billion tons, which accounts for 50.5% of the total global production, more than half. China's current energy mix also makes its environmental problems more acute as coal is difficult to extract and use without emitting carbon dioxide. However, environment is one of the evaluation criteria for the quality of economic growth [4], so the coordinated development of environment and economy has become an important goal pursued by the Chinese government. In June 2007, the Chinese government formulated China's National Climate

Change Program and began to tackle the challenge of climate change head-on. Since the 18th CPC National Congress, policy concepts such as "ecological civilization construction", "ecological environmental protection", "low-carbon economy", and "emission reduction" have been gradually formed, and policy actions in the field of environmental protection have been gradually implemented. In recent years, the Chinese government has taken carbon emission reduction as an important breakthrough in environmental protection and has begun to develop clean energy, work to electrify transportation, and promote zero-carbon buildings vigorously, etc. At the UN General Assembly in September 2020, the Chinese government promised that "China will enhance its nationally determined contribution and adopt more effective policies and measures. China would strive to achieve carbon peak by 2030 and achieve carbon neutrality by 2060" [5]. According to data released by the National Bureau of Statistics, the share of coal consumption in China's total energy consumption has been declining year by year and has dropped by more than 10 percent in the last decade. However, as the largest developing country and the second largest economy in the world, China needs to achieve economic growth while attaching importance to environmental protection. Green finance provides a reliable tool to help China achieve carbon neutrality.

Based on the importance of transforming a "grey economy" into a "green economy", the Chinese government began to develop green finance in 2007, which directly led to a significant decrease in new bank loans of "high polluting and energy consuming" enterprises and a significant decrease in urban sulfur dioxide emissions and industrial wastewater emissions [6]. China first proposed building a green financial system in 2015, marking that the Chinese government began to attach importance to developing green finance. In the 13th Five-Year Plan, the concept of "green development" was introduced for the first time in China to "establish a green financial system, develop green credit, green bonds and set up a green development foundation". The report of the 19th National Congress of the Communist Party of China (CPC) emphasizes "building a market-oriented green technology innovation system, developing green finance, and growing the energy conservation and environmental protection industry, clean production industry, and clean energy industry" [7]. In 2019, the National Development and Reform Commission and another seven ministries and commissions jointly issued the Green Industry Guidance Catalogue (2019 Edition), which proposes the development priorities of green industry, covering six categories, including energy conservation and environmental protection, clean production, clean energy, ecological environment, green infrastructure upgrading, and green services. China's green finance policy system is heading for a more mature and diversified development period. According to the data released by the Ministry of Ecology and Environment of China, as of June 2013, the balance of green credit in China is only 4.85 trillion yuan, but, by the end of 2021, it increased to 15.9 trillion yuan, ranking first in the world. Before 2015, the issuance scale of green bonds in China was almost zero, but, by 2021, it reached 440.1 billion yuan, which is the second largest green bond market in the world from the perspective of cumulative issuance and annual issuance. It can be said that the green bond market started from zero and developed rapidly. Similarly, investment in green projects was almost zero in 2015 and before, but it reached 5.3729 trillion yuan by 2020. It can be seen that China attaches great importance to the coordinated development of green finance and environmental protection. Therefore, although green finance started late in China, it has developed rapidly, and it is necessary, important, and urgent to study how to make green finance better serve environmental protection.

At present, scholars have studied China's environmental protection issues from the perspectives of the government [8,9], enterprises [10], and individuals [11]. Studies on green finance are also emerging, mainly focusing on policy suggestions for the development of green finance and its multifaceted functions and influences [12–14], but there is no research on the coordination between environmental protection and green finance. In recent years, China has an increasingly urgent demand for environmental protection and pays more attention to the development of green finance. However, green finance started late in China, and whether it can be well coordinated and integrated with China's

environmental protection goals and policies remains to be tested, and how to improve the promoting effect of green finance on environmental protection also remains to be studied. Currently, most scholars at home and abroad study the interaction among systems by measuring the "coupling and coordinated degree" [15,16], which mainly shows the overall changing evolution process from low to high, from easy to complex, and from chaos to integration among various elements of the system and the system itself [17]. Wang et al. (2022) verify the contribution of green finance to carbon neutrality in China by using correlation tests, GM (1,1), and the BP neural network model to predict the expected time of carbon peak and carbon neutrality in China under the effect of green finance. This paper extends the research object to the whole environmental protection system and selects four scientific and representative relevant indicators of the green finance system and environmental protection system, respectively, based on a great deal of existing scientific research. Meanwhile, this paper subdivides the research into 30 provinces and divides them into four regions: the Eastern, Central, Western, and Northeast regions. In addition to the national scope, this paper also studies, compares, and analyzes the coupling and coordinated level of each province and each region, respectively. What is more, it does not only study the simple correlation between green finance and environmental protection but also studies the multiple and complex relationships between the two systems and the fit degree. Further, the improved GM (1,1) model based on background value optimization is used to predict whether and when the coupling and coordinated level of the GE system can reach the coordination level nationwide and in each region. First of all, this paper adopts the "coupling and coordinated degree" to quantify the interaction and fit level between green finance and environmental protection in each province based on the relevant data of 30 provinces from 2011 to 2020 and calculates the average level in each region and nationwide. Then, the improved GM (1,1) model based on background value optimization is used to predict the changing trend of the average coupling and coordinated level of the GE system nationwide and in each region in the future and whether or when they will reach the coordination level. Finally, based on the research results, this paper puts forward scientific policy suggestions to improve the coupling and coordinated level between domestic green finance development and environmental protection policies. This paper aims to study how to integrate green finance development and environmental protection policies more effectively nationwide and in different regions so as to make green finance better serve environmental protection objectives, improve the coupling and coordinated level of the GE system, and accelerate the process of green development and the transformation from a "grey economy" to a "green economy" in China.

## 2. Literature Review

### 2.1. Development and Research Status of Green Finance in China

Green finance is inseparable from green development and environmental protection. Green finance is an important part of China's financial reform and innovation, as well as a powerful means to realize China's economic structural transformation. China's green finance policy framework is guided to establish and improve the green financial system, which stops new credit support for projects with high carbon emissions and high pollution while launching innovative green financial products to encourage capital demand for energy conservation and environmental protection projects.

The concept of "green finance" in China is not fundamentally different from the international concept of "environmental finance". The basic idea of "environmental finance" is to solve the environmental problems that human beings are facing through financial instruments, and "green finance" largely comes from spontaneous demand for financial service in the process of economic structure transformation to a low-carbon economy [18]. China started to develop green finance formally in 2015, and it is still in the exploration stage, with many deficiencies and limitations to be improved. Some scholars suggest increasing investment and financing of green technology and green infrastructure and using green finance as a breakthrough point to promote diversified investment [19]. Financial institutions and en-

terprises are the important subjects of green investment and financing, and they can guide social capital into green industries, such as energy conservation, environmental protection, and green manufacturing, through credit, bonds, insurance, and other related products to support the upgrading of green technology and effectively promote the development of green finance [20]. Accordingly, green financing can also alleviate tightness of enterprises' green capital and regulate the awareness of enterprises' ecological innovation. Good and effectively coordinated polices between green finance and environmental protection will promote enterprises' ecological innovation [21]. Furthermore, studies have shown that corporations' participation in environmental responsibility practice in the context of green financial development not only increases shareholder value but also increases the value of non-financial stakeholders [22]. In recent years, the Chinese government has injected a large number of investment funds into ESG (environment, social, and governance) activities based on the impact of green finance on environmental protection achievements and has effectively attracted private investors to conduct similar capital allocation to provide green financing for low-carbon energy and other green innovations to promote the process of environmental protection and sustainable development [23].

### 2.2. Progress and Research Status of Environmental Protection in China

In the face of global environmental problems and China's own environmental protection dilemma, China has been closely following the pace of global environmental governance, making Chinese contributions by constantly improving environmental protection practices and committing to introducing more effective policies to increase investment in environmental protection, as well as introducing the concept of environmental protection into all walks of life.

The situation varies from region to region in terms of the economy, resources, environment, and other aspects. Economically underdeveloped regions with weak financial self-sufficiency need to increase investment in environmental protection, while economically developed regions should pay more attention to environmental supervision [24] in order to narrow regional differences and promote the overall process of environmental protection in China. Some scholars have studied the environmental protection tax in China, suggesting that the green tax system should be continuously expanded in terms of the tax categories included and combining tax and charging policies to complete the green tax system [25]. It is widely believed by the public that "climate policy is expensive", which often leads to low efficiency and weak efforts in the formulation of environmental protection policies. However, studies have proven that strict climate policy is much less costly than expected and is conducive to deep decarbonization [26]. Enterprises' decisions and actions play important roles in economic, social, and environmental protection, but the cost–benefit mismatch caused by environmental regulations severely impacts the incentive of enterprises to protect the environment. To realize a win–win situation for environmental protection and business development, enterprises must release information about environmental planning to obtain investors' support first of all [27]. In addition to enterprises, public participants' pro-environment behavior and socially responsible investment are also important factors affecting the process of environmental protection. However, public participants are rarely intentionally pro-environment or make environmental protection financial investments in real life, so Walczak Damian and other scholars suggest that social participation can be increased when formulating environmental policies to enhance public awareness of environmental protection through environmental protection activities and reasonable mandatory policies [28].

### 2.3. Research Status of Coupling and Coordinated among Systems

The concept of "coupling" originally comes from physics and is defined as the phenomenon that systems or elements influence each other because of interaction and even move synergically [29]. At present, scholars mostly study the complex relationship between systems through "coupling" and quantify the interaction relationship and fit level among

systems by measuring the "coupling and coordinated degree", which is widely used. Wang and Han (2021) use "coupling" to study the characteristics of coordinated development level, space-time evolution law, and influencing factors of urban economy, society, and environment in China [30]. Lv and Cui (2020) measure the coupling and coordinated degree between global value chain embedding and ecological environment under different types of value chain drivers and study whether and how they can achieve a win–win situation [31]. In addition, by constructing a climate–society coupling system, an interdisciplinary and cross-system feedback process can be studied, the interaction between multiple systems can be quantified, and multidimensional, comprehensive, specific, and scientific policies and emission paths can be analyzed [32]. By measuring the coupling and coordinated degree between urban development and ecological environment, the potential relationship between urban development level and its sustainability can be analyzed [33]. Some scholars also apply "coupling" to study the relationship between the development of the medical service industry and pharmaceutical manufacturing industry, analyzing the influencing factors and their spatial spillover effect [34]. Nowadays, the method of system coupling has been maturely applied in all aspects and fields, which confirms the scientific rationality of this method.

## 3. Source of Data and Methods of Research

### 3.1. Source of Data

In order to better measure coupling and coordinated level of China's green finance–environmental protection system, this study combines a large amount of relevant and high-quality scientific academic research results and selects eight indexes for green finance and environmental protection. Among them, referring to research by Li et al. (2021) on coupling and coordinated mechanism of green technology innovation, environmental regulation, and green finance, this paper selects the A-share market value of energy-saving and environmental protection enterprise and the completed investment in treatment of industrial pollution this year as the index of green financial system and the harmless treatment volume of domestic waste as the index of environmental protection system [35]. According to the study by Sun et al. (2021) on the coupling between green technological innovation and the development of green financial system, the issuing scale of green bonds and the investment of environmental health of municipal public facilities in organized towns are selected as the indicators of the green financial system in this paper [36]. Referring to the study by Wei et al. (2021) on the coupling and coordinated relationship between ecological sustainability and high-quality economic development, the carbon dioxide emissions, forest coverage rate, and sulfur dioxide emissions are adopted as the indicators of the environmental protection system in this paper [37]. Combining the above-mentioned scholars' studies with the reference to this study, the indicators selected in this paper are scientific, representative, and reasonable.

China developed green finance formally in 2015, and, after 2010, the concepts of environmental protection and carbon neutrality began to receive more and more attention globally, as in China, and China committed to the time of China's carbon peak and carbon neutrality at the United Nations General Assembly in 2019. Therefore, the data related to environmental protection and green finance in the last decade are more representative and meaningful for research. Based on the above objective reasons, this paper selects panel data of 30 provinces in mainland China from 2011 to 2020, which are taken as samples. Considering the availability of data, data from Tibet are not included in the investigation. The data mainly come from WIND database, China Statistical Yearbook on environment, and China Statistical Yearbook. The specific indicators are as follows (Table 1):

**Table 1.** Evaluation index system of green finance–environmental protection system (GE system) coupling and coordinated level.

| First-Level Indicators | Second-Level Indicators | Unit | Index Attribute |
|---|---|---|---|
| Green finance | the issuing scale of green bonds | One hundred million yuan | + |
| | the completed investment in treatment of industrial pollution this year | Ten thousand yuan | + |
| | A-share market value of energy-saving and environmental protection enterprise | yuan | + |
| | the investment of environmental health of municipal public facilities in organized towns | Ten thousand yuan | + |
| Environmental protection | carbon dioxide emissions | Thousand ton | - |
| | forest coverage rate | % | + |
| | the harmless treatment volume of domestic waste | Ten thousand ton | + |
| | sulfur dioxide emissions | Ten thousand ton | - |

Since China started to develop green finance in 2015, the data collected on the issuing scale of green bonds in 2015 and before are all zero, and, in 2016, only Beijing, Jiangsu, Zhejiang, and Hubei had issued green bonds, indicating that these provinces started to develop green finance earlier. Afterwards, other provinces started to develop this green financial product and gradually increase the scale. The overall completed investment in treatment of industrial pollution this year in all 30 provinces has been decreasing year by year, which shows that industrial pollution in China has been reduced in the last decade and the environmental protection policy is beginning to bear fruit. The overall domestic A-share market value of energy-saving and environmental protection enterprises has been rising year by year, and the provinces with energy-saving and environmental protection enterprises have also basically maintained an upward trend, but it is worth noting that, from 2011 to 2020, this index in nine provinces, including Shanxi, Liaoning, Yunnan, and Xinjiang, is 0. This is inextricably linked to the local environmental protection process. In addition, most provinces show a trend of year-on-year increase in the investment in environmental health of municipal public facilities in organized towns, but there are individual provinces that show a trend of increase followed by decrease, such as Beijing and Tianjin, which may be related to local urbanization policies. Although carbon dioxide emissions and sulfur dioxide emissions are both polluting gases, the panel data show that carbon dioxide emissions in 30 provinces are still increasing year by year, which is closely related to China's coal-based energy structure, while sulfur dioxide emissions as a whole are decreasing year by year, which also indicates that the use of sulfur-dioxide-generating energy has been reduced and China is trying to carry out energy transformation. The forest coverage rate and the harmless treatment volume of domestic waste in all provinces show a year-on-year decrease, and these provinces all respond positively to the national environmental protection policy and the goal of carbon emission reduction. The panel data of the eight domestic indicators selected in this paper are all characterized by changes, and the differences among the provinces can be seen through observation and analysis, and these indicators and data reflect the current situation of environmental protection and green finance development in China to a certain extent, which is indicative and representative.

*3.2. Methods of Research*

3.2.1. Method of Measuring the Coupling and Coordinated Level of GE System

On the basis of determined indicator system, this paper uses the range method, entropy method, coupling coordination degree model, and other methods and formulas in turn [35,38] to calculate the coupling and coordinated degree level of GE system. The results of the weight of indicators at the coupling and coordinated level are shown in Table 2. By collecting samples and data, there are m samples and n indicators.

**Table 2.** Weight of indicators at the coupling and coordinated level of green finance and environmental protection system (GE system).

| Second-Level Indicators | Average Weight |
|---|---|
| the issuing scale of green bonds | 0.2956 |
| the completed investment in treatment of industrial pollution this year | 0.1226 |
| A-share market value of energy-saving and environmental protection enterprise | 0.2840 |
| the investment of environmental health of municipal public facilities in organized towns | 0.1929 |
| carbon dioxide emissions | 0.0365 |
| forest coverage rate | 0.0771 |
| the harmless treatment volume of domestic waste | 0.1159 |
| sulfur dioxide emissions | 0.0399 |

1.  Range method.

Since the dimensions of each indicator are different, the range method is used to standardize the initial data to eliminate the dimensionality effect before data analysis. The standardized calculation method is:

$$x'_{ij} = \begin{cases} \frac{x_{ij} - \min x_j}{\max x_j - \min x_j}, & \text{Standardization of positive iindicators} \\ \frac{\max x_j - x_{ij}}{\max x_j - \min x_j}, & \text{Standardization of negative indicators} \end{cases} \quad (1)$$

In the above formula, $x'_{ij}$ represents the normalized result, $x_{ij}$ represents the original value of the $j$th index of the $i^{th}$ sample ($1 \leq i \leq m$, $1 \leq j \leq n$), $\min x_j$ represents the minimum value of the $j$th item index, and $\max x_j$ represents the maximum value of the $j$th item index. Normalization of standardized index values:

$$y_{ij} = x'_{ij} \cdot 0.99 + 0.01 \quad (2)$$

After standardizing all sample data from 30 provinces of China from 2011 to 2020, the weight of 8 indicators can be further calculated and analyzed by using range method.

2.  Entropy method.

The entropy method is used to calculate the proportion of the $j$th index of the $i^{th}$ sample in the sum of the index $Q_{ij}$:

$$Q_{ij} = y_{ij} / \sum_{i=1}^{m} y_{ij} \quad (3)$$

Then, calculate the entropy value $e_j$ of this indicator; let

$$e_j = -\frac{1}{\ln m} \sum_{i=1}^{m} p_{ij} \cdot \ln p_{ij} \quad (4)$$

$m$ be the sample size
Calculate the difference coefficient $g_j$ of the JTH index:

$$g_j = 1 - e_j \quad (5)$$

The larger the difference coefficient value is, the greater the influence of the index in the comprehensive evaluation of the system is. Thus, the weight of each indicator $w_j$:

$$w_j = g_j / \sum_{j=1}^{n} g_j \quad (6)$$

3.  Coupling coordination degree model.

Referring to the research of Li and Cui (2018) [39] and based on the capacity coupling concept and capacity system model in physics, in order to measure the degree of interaction

between systems more accurately, this paper uses the coupling coordination degree model to judge the synergy of GE system:

$$C(U_1, U_2, \ldots U_n) = n \times \left[ \frac{U_1 U_2 \ldots U_n}{(U_1 + U_2 + \cdots + U_n)^n} \right]^{1/n} \tag{7}$$

Thus, the coupling degree model of green finance–environmental protection system can be obtained:

$$C(U_1, U_2) = n \times \left[ \frac{U_1 U_2}{(U_1 + U_2)^2} \right]^{1/2} \tag{8}$$

The comprehensive score of each system in each province is calculated by the following formula:

$$U_{Ni} = \sum_{j=1}^{n} w_j \cdot y_{ij} \tag{9}$$

Although the coupling degree model can reflect the degree of interaction between multiple systems, it cannot judge the environment in which the composite system is developing. There may be a situation where the overall level of the GE system is low, i.e., the overall is in a poorly coordinated environment, but its coupling degree is high instead. The degree of coupling coordination refers to the degree of benign interaction among various systems and is an important indicator to judge the overall development level. Therefore, in order to more accurately understand the coordination development situation of GE system, the coupling coordination degree model needs to be introduced:

$$D = \sqrt{C \times T} \tag{10}$$

Here, $T = \alpha U_1 + \beta U_2$, $D$ denotes the coupling coordination degree of the composite system, and $T$ denotes the comprehensive coordination index of each system, which can reflect their contribution to the overall coordinated development. Further, $\alpha$ and $\beta$ are the weight coefficients of each system. Considering that green finance and environmental protection have equally important influence on the overall development, $\alpha = \beta = 1/2$. The coupling coordination degree $D$ takes the value in the range of 0–1. Based on previous relevant studies, the classification standard of coupling and coordinated degree is shown in the following Table 3:

**Table 3.** Classification standard of coupling and coordinated degree.

| Coupling and Coordinated Degree $D$ | Coupling and Coordinated Degree Level | Range | Classification |
|---|---|---|---|
| [0, 0.1) | Extreme disharmony | | |
| [0.1, 0.2) | Severe disharmony | | |
| [0.2, 0.3) | Moderate disharmony | $0 \ll D < 0.4$ | Disharmonized decline |
| [0.3, 0.4) | Mild disharmony | | |
| [0.4, 0.5) | On the verge of disharmony | | |
| [0.5, 0.6) | Barely coordination | $0.4 \ll D < 0.6$ | Intermediate transition |
| [0.6, 0.7) | Primary coordination | | |
| [0.7, 0.8) | Intermediate coordination | | |
| [0.8, 0.9) | good coordination | $0.6 \ll D \ll 1$ | Coordinated lifting |
| [0.9, 1) | High-quality coordination | | |

### 3.2.2. Prediction of Coupling and Coordinated Level of GE System

Based on the coupling and coordinated degree of GE system from 2011 to 2020, this paper predicts the average level of coupling and coordinated degree of GE system in four regions and nationwide in the next 60 years. By using MATLAB software, the grey prediction GM (1,1) model, GM (1,1) residual model, and the improved GM (1,1) model based on background value optimization are established. Finally, the improved GM (1,1) model based on background value optimization with the highest prediction accuracy was selected. Through the prediction, study when the average coupling and coordinated

degree of GE system can achieve coordination under the current situation of nationwide green financial development and environmental protection, as well as the slow growth of coupling and coordinated degree. Then, according to the forecast results, this paper proposes timely, efficient, and powerful policy suggestions and provides reference for the Chinese government and related departments.

Grey system refers to the system containing both known information and unknown or uncertain information; it predicts grey processes that are time-dependent and vary within a certain range. Grey system theory was first proposed by Deng (1985) [40], and it solves the problem of modeling continuous differential equations, which have been considered unsolvable. Further, grey system theory has been applied to many sectors and multiple fields, such as population, economy, ecology, medicine, agriculture, hydrology, and disaster mitigation, through continuous improvement and development [41]. GM (1,1) model is suitable for predicting regular data changes, and, according to the characteristics of coupling and coordinated degree results of the last ten years nationwide and regionally measured in this paper, they all belong to regular monotonic change. In addition, the grey prediction GM (1,1) model does not require a large number of data samples, so it is suitable for the ten-year sample data selected in this paper. GM (1,1) model is the most basic model of grey theory. The modeling principle is simple and easy to operate, but there are inherent defects. The GM (1,1) model optimized based on background value is chosen in this paper as an improvement of the traditional GM (1,1) model, which improves the applicability and prediction accuracy of the traditional GM (1,1) model [42]. This method has been recognized by a large number of scientific and high-quality literature studies and widely used. Jiang et al. (2014) optimized the background value of GM (1,1) model from the perspective of integral geometric meaning by using the idea of function approximation. The results show that the GM (1,1) model established by the optimized background value calculation formula can significantly improve the prediction accuracy [43]. Jiang et al. (2015) believe that it was an effective method to improve GM (1,1) model by optimizing the background value. Based on the core idea of Riemann integral, they optimize the background value and verify the reliability and practicability of the improved GM (1,1) model [44]. Zu et al. (2018) use GM (1,1) model based on background value optimization to predict the future GDP data of Mudanjiang City and compare the prediction error of GM (1,1) model based on background value optimization with that of traditional GM (1,1) model and find that the former has better prediction accuracy than the latter [45]. There are many methods for background value optimization, which can be summarized as numerical integration, weighted background value, and background value construction based on discrete exponential function [46]. The method of background value optimization adopted in this paper is numerical integration, and the specific modeling process is as follows:

Step 1: Set a set of raw non-negative data:

$$x^{(0)} = (x^{(0)}(1), x^{(0)}(2), \ldots, x^{(0)}(n)) \tag{11}$$

$n$ is the number of data. Accumulate $x^{(0)}$ to weaken the volatility and randomness of the random sequence; a new sequence is derived:

$$x^{(1)} = (x^{(1)}(1), x^{(1)}(2), \ldots, x^{(1)}(n)) \tag{12}$$

$$x^{(1)}(k) = \sum_{i=1}^{k} x^{(0)}(i), k = i, 2, \ldots, n \tag{13}$$

Step 2: Establish the differential equation of gray whitening for $x^{(1)}$:

$$\text{GM}(1,1): \frac{dx^{(1)}}{dt} + ax^{(1)} = u \tag{14}$$

$a$ and $u$ are the coefficients to be solved, which, respectively, become the development coefficient and the gray action, and integrate the above equation over the interval $[k, k + 1]$ and obtain:

$$x^{(1)}(k+1) - x^{(1)}(k) + a \int_k^{k+1} x^{(1)}(t)dt = u, k = 1, 2, \ldots, n-1 \tag{15}$$

That is:

$$x^{(0)}(k+1) + a \int_k^{k+1} x^{(1)}(t)dt = u \tag{16}$$

Step 3: Let

$$Z^{(1)}(k+1) = \int_k^{k+1} x^{(1)}(t)dt \tag{17}$$

be the background value of $x^{(1)}(t)$ on the interval $[k, k + 1]$; then:

$$x^{(0)}(k+1) + aZ^{(1)}(k+1) = u \tag{18}$$

Since the solution of the first-order differential equation is an exponential function, let

$$x^{(1)}(t) = ce^{bt} \tag{19}$$

And assuming the curve crossing point $(k, x^{(0)}(k))$ and $(k+1, x^{(0)}(k+1))$, then:

$$x^{(1)}(k) = ce^{bk} \tag{20}$$

$$x^{(1)}(k+1) = ce^{b(k+1)} \tag{21}$$

It can be obtained from (19) and (20):

$$b = lnx^{(1)}(k+1) - lnx^{(1)}(k), c = \frac{x^{(1)}(k)}{e^{bk}} = \frac{[x^{(1)}(k)]^{k+1}}{[x^{(1)}(k+1)]^k} \tag{22}$$

Therefore, the background value is:

$$Z^{(1)}(k+1) = \int_k^{k+1} x^{(1)}(t)dt = \int_k^{k+1} ce^{bt}dt = \frac{x^{(1)}(k+1) - x^{(1)}(k)}{lnx^{(1)}(k+1) - lnx^{(1)}(k)} \tag{23}$$

Step 4: Use the least square method to solve $a$ and $u$,

$$\hat{\oslash} = [\hat{a}\hat{u}]^T = \left(B^T B\right)^{-1} B^T \tag{24}$$

$$B = \begin{bmatrix} -Z^{(0)}(2) & 1 \\ -Z^{(0)}(3) & 1 \\ \ldots & \ldots \\ -Z^{(0)}(n) & 1 \end{bmatrix}, Yn = \begin{bmatrix} x^{(0)}(2) \\ x^{(0)}(3) \\ \ldots \\ x^{(0)}(n) \end{bmatrix} \tag{25}$$

After solving $a$ and $u$, the discrete solution of the whitening equation can be obtained as follows:

$$\hat{x}^{(1)}(k+1) = \left(x^{(1)}(1) - \frac{\hat{u}}{\hat{a}}\right)e^{-ak} + \frac{\hat{u}}{\hat{a}} \tag{26}$$

Accumulate the above data inversely and recover the data; the simulated value of the original data sequence can be obtained as follows:

$$\hat{x}^{(0)}(k+1) = \hat{x}^{(1)}(k+1) - \hat{x}^{(1)}(k) = (1 - e^a)[x^{(1)}(1) - \frac{\hat{u}}{\hat{a}}]e^{-\hat{a}k} \tag{27}$$

This study tests the accuracy of the established improved GM (1,1) model based on background value optimization. The prediction accuracy level and comparison table of index tests are cited from Wang et al. (2022) [14], as shown in Table 4:

**Table 4.** Comparison table of prediction accuracy.

| Level | Variance Ratio (C) | Small Probability Error (P) | Correlation (R) | Relative_Error_Mean | |
|---|---|---|---|---|---|
| Good | C ≤ 0.35 | P ≥ 0.95 | R > 0.9 | Accuracy level | |
| qualified | 0.35 < C ≤ 0.5 | 0.95 > P ≥ 0.8 | 0.9 ≥ R > 0.8 | excellent | <0.01 |
| Barely qualified | 0.5 < C ≤ 0.65 | 0.8 > P ≥ 0.7 | 0.8 ≥ R > 0.7 | qualified | 0.01–0.05 |
| unqualified | C > 0.65 | P < 0.7 | 0.7 ≥ R > 0.6 (satisfied) | Barely qualified | 0.05–0.1 |

## 4. Results of Research

### 4.1. Measurement Results of GE System Coupling and Coordinated Level

The sample data were standardized by the range method, and then the weight of each indicator was calculated by the entropy method, and the coupling coordination degree model was used to calculate the coupling and coordinated-degree-related data of the GE system in 30 provinces in China from 2011 to 2020. The results are as follows:

According to the data in Table 5, the average coupling and coordinated degree of the GE system in 30 provinces in China from 2011 to 2020 is mildly disharmonized; that is, the overall level is not high nationwide. Among the 30 provinces, only Zhejiang and Guangdong's average coupling and coordinated degree of the GE system achieved coordination between 2011 and 2020, while all the rest did not. Meanwhile, the 30 provinces' changing trends and characteristics are very different; for instance, Guangdong's coupling and coordinated degree shows a trend of growing and grows significantly in the past decade, Ningxia Hui Autonomous Region has little fluctuation and a gentle trend, while Inner Mongolia Autonomous Region even shows a gradual declining trend.

**Table 5.** Calculation results of coupling and coordinated degree of 30 provinces and in China from 2011 to 2020.

| | 2011 | 2012 | 2013 | 2014 | 2015 | 2016 | 2017 | 2018 | 2019 | 2020 | Mean |
|---|---|---|---|---|---|---|---|---|---|---|---|
| Beijing | 0.4699 | 0.4703 | 0.4773 | 0.4635 | 0.4122 | 0.4666 | 0.4650 | 0.4933 | 0.3715 | 0.5280 | 0.4630 |
| Tianjin | 0.3235 | 0.3118 | 0.3058 | 0.3118 | 0.3431 | 0.2988 | 0.2950 | 0.2794 | 0.3412 | 0.3828 | 0.3225 |
| Hebei | 0.2840 | 0.2953 | 0.3109 | 0.3030 | 0.3261 | 0.2849 | 0.2908 | 0.3565 | 0.2840 | 0.3039 | 0.3013 |
| Shanxi | 0.2540 | 0.2709 | 0.2953 | 0.2377 | 0.2413 | 0.2426 | 0.3111 | 0.2621 | 0.2645 | 0.2812 | 0.2660 |
| Inner Mongolia | 0.3064 | 0.3213 | 0.3500 | 0.3209 | 0.3092 | 0.2988 | 0.2830 | 0.2758 | 0.2390 | 0.2631 | 0.2978 |
| Liaoning | 0.2897 | 0.3116 | 0.3185 | 0.2876 | 0.2712 | 0.2669 | 0.2532 | 0.2338 | 0.2317 | 0.2650 | 0.2750 |
| Jilin | 0.2559 | 0.2484 | 0.2500 | 0.2817 | 0.2778 | 0.2581 | 0.2587 | 0.2556 | 0.2641 | 0.2531 | 0.2616 |
| Heilongjiang | 0.3176 | 0.2989 | 0.3657 | 0.3408 | 0.3342 | 0.3048 | 0.3132 | 0.3063 | 0.2988 | 0.2903 | 0.3114 |
| Shanghai | 0.3920 | 0.3401 | 0.3438 | 0.3154 | 0.3310 | 0.3542 | 0.3736 | 0.3526 | 0.3598 | 0.3611 | 0.3503 |
| Jiangsu | 0.4966 | 0.5332 | 0.4703 | 0.4750 | 0.4888 | 0.4929 | 0.4948 | 0.4934 | 0.4021 | 0.5124 | 0.4857 |
| Zhejiang | 0.5450 | 0.5541 | 0.5989 | 0.5443 | 0.5428 | 0.5557 | 0.5250 | 0.5513 | 0.5229 | 0.5698 | 0.5486 |
| Anhui | 0.3256 | 0.3199 | 0.3861 | 0.2909 | 0.2972 | 0.3185 | 0.3290 | 0.3499 | 0.3363 | 0.3617 | 0.3309 |
| Fujian | 0.3609 | 0.3799 | 0.3944 | 0.3731 | 0.4090 | 0.3711 | 0.3819 | 0.3822 | 0.3760 | 0.4287 | 0.3853 |
| Jiangxi | 0.3100 | 0.2923 | 0.3204 | 0.2904 | 0.3007 | 0.3033 | 0.3405 | 0.3203 | 0.3311 | 0.3543 | 0.3150 |
| Shandong | 0.3966 | 0.4007 | 0.3420 | 0.3997 | 0.4238 | 0.4159 | 0.4131 | 0.4185 | 0.3703 | 0.4956 | 0.4066 |
| Henan | 0.3241 | 0.3165 | 0.3687 | 0.3203 | 0.3211 | 0.3487 | 0.3718 | 0.3695 | 0.3549 | 0.4392 | 0.3502 |
| Hubei | 0.3490 | 0.3533 | 0.3957 | 0.3713 | 0.3584 | 0.4894 | 0.3773 | 0.4529 | 0.4566 | 0.3800 | 0.3976 |
| Hunan | 0.3154 | 0.3591 | 0.3604 | 0.3372 | 0.3479 | 0.3366 | 0.3624 | 0.3489 | 0.3486 | 0.3909 | 0.3472 |
| Guangdong | 0.5094 | 0.5355 | 0.4981 | 0.5478 | 0.5572 | 0.5472 | 0.5793 | 0.6116 | 0.5806 | 0.6418 | 0.5582 |
| Guangxi | 0.2527 | 0.2559 | 0.2824 | 0.2634 | 0.2988 | 0.2832 | 0.2971 | 0.2920 | 0.3005 | 0.2884 | 0.2795 |
| Hainan | 0.2035 | 0.2088 | 0.1925 | 0.1989 | 0.2079 | 0.2192 | 0.2287 | 0.2262 | 0.2292 | 0.2988 | 0.2182 |
| Chongqing | 0.2582 | 0.3046 | 0.2985 | 0.2883 | 0.3067 | 0.2948 | 0.3535 | 0.3168 | 0.3053 | 0.3640 | 0.3098 |
| Sichuan | 0.4274 | 0.4184 | 0.4631 | 0.4342 | 0.4098 | 0.3847 | 0.3942 | 0.4944 | 0.4065 | 0.4142 | 0.4231 |
| Guizhou | 0.2474 | 0.2393 | 0.2581 | 0.2461 | 0.2489 | 0.2609 | 0.2648 | 0.2964 | 0.2978 | 0.3444 | 0.2669 |
| Yunnan | 0.2597 | 0.2755 | 0.2734 | 0.2453 | 0.2673 | 0.2497 | 0.2614 | 0.3428 | 0.3162 | 0.3229 | 0.2802 |
| Shanxi | 0.3776 | 0.3803 | 0.3841 | 0.3530 | 0.3725 | 0.3387 | 0.3465 | 0.3550 | 0.3571 | 0.3416 | 0.3632 |
| Gansu | 0.2052 | 0.2393 | 0.2153 | 0.1895 | 0.1707 | 0.2035 | 0.2001 | 0.2111 | 0.2435 | 0.2096 | 0.2116 |
| Qinghai | 0.1671 | 0.1590 | 0.1461 | 0.1577 | 0.1688 | 0.1893 | 0.1616 | 0.1716 | 0.1982 | 0.1590 | 0.1685 |
| Ningxia | 0.1748 | 0.1889 | 0.2141 | 0.2199 | 0.2004 | 0.2263 | 0.1929 | 0.2012 | 0.2052 | 0.2010 | 0.2019 |
| Xinjiang | 0.2047 | 0.1868 | 0.2207 | 0.2035 | 0.1985 | 0.2013 | 0.1952 | 0.2304 | 0.1926 | 0.1964 | 0.2027 |
| Mean | 0.3201 | 0.3257 | 0.3367 | 0.3204 | 0.3248 | 0.3269 | 0.3305 | 0.3417 | 0.3262 | 0.3548 | |
| Level | Mild disharmony | Mild disharmony | Mild disharmony | Mild disharmony | Mild disharmony | Mild disharmony | Mild disharmony | Mild disharmony | Mild disharmony | Mild disharmony | |

In this paper, the differences and characteristics of regions are analyzed by comparing the coupling and coordinated degree of the GE system in the Eastern region, Western region, Central region, and Northeast region (as shown in Figure 1).

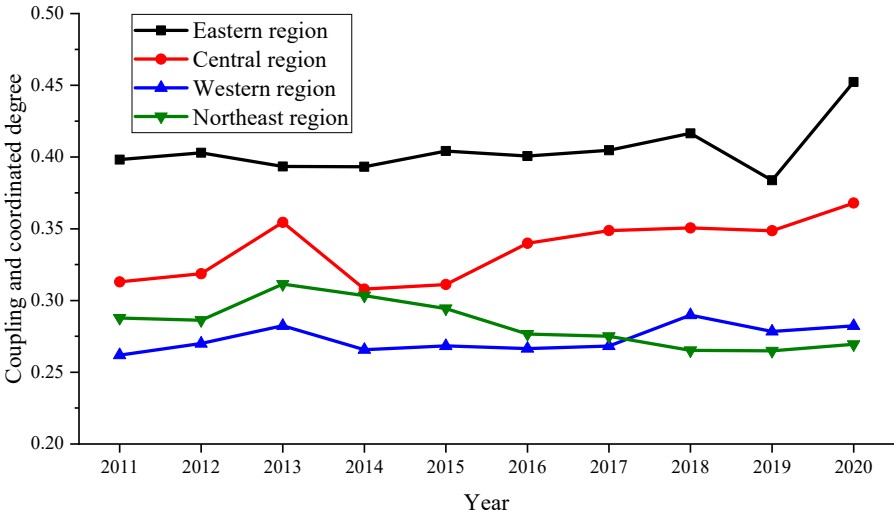

**Figure 1.** Variation trend in GE system coupling coordination degree in Eastern, Central, Western, and Northeast regions.

According to the level and change trend of the coupling and coordinated degree of the GE system in the four regions of China in Figure 2, it can be concluded that: (1) the overall coupling and coordinated degree in the Eastern region has been in the range of 0.4–0.5 in the recent ten years with a slow growth, and, after a significant decline in 2019, it shows a rapid growth in 2020, ranking first among the four regions. (2) The coupling and coordinated level of the GE system in the Central region is basically in the range of 0.3–0.4, showing a characteristic of fluctuating growth, and continues to increase after declining in 2014, ranking second among the four regions. (3) The Northeast region basically remains at 0.25–0.3, which is better than that in the Western region before 2016. However, it gradually declines in 2016 and later, and begins to be equal to that in the Western region. (4) The Western region also remains at 0.25–0.3 in the recent ten years with small fluctuations, a gentle trend, and stagnation. (5) From a national perspective, except for the Northeast region, which shows a downward trend, the coupling and coordinated degrees in the other three regions all demonstrate a slow upward trend.

*4.2. Prediction Results of GE System Coupling and Coordinated Level*

The accuracy test results of the GM (1,1) model based on background value optimization constructed for the prediction of average coupling and coordinated degree in the Eastern region, Central region, Western region, Northeast region and nationwide in this paper are shown in Table 6:

**Table 6.** The accuracy test results of regions and nationwide.

|  | Eastern Region | | Central Region | | Western Region | | Northeast Region | | Nationwide | |
|---|---|---|---|---|---|---|---|---|---|---|
| **Test Index** | **Value** | **Result** | **Value** | **Result** | **Value** | **Result** | **Value** | **Result** | **Value** | **Result** |
| C | 0.5562 | Barely qualified | 0.4130 | qualified | 0.4801 | qualified | 0.2927 | qualified | 0.3796 | qualified |
| P | 0.8 | qualified | 0.900 | qualified | 0.8 | qualified | 1 | good | 0.9 | qualified |
| R | 0.7547 | satisfied | 0.6853 | satisfied | 0.6235 | satisfied | 0.7122 | satisfied | 0.7096 | satisfied |
| Rel_Error_Mean | 0.0240 | qualified | 0.0288 | qualified | 0.0198 | qualified | 0.0255 | qualified | 0.0198 | qualified |

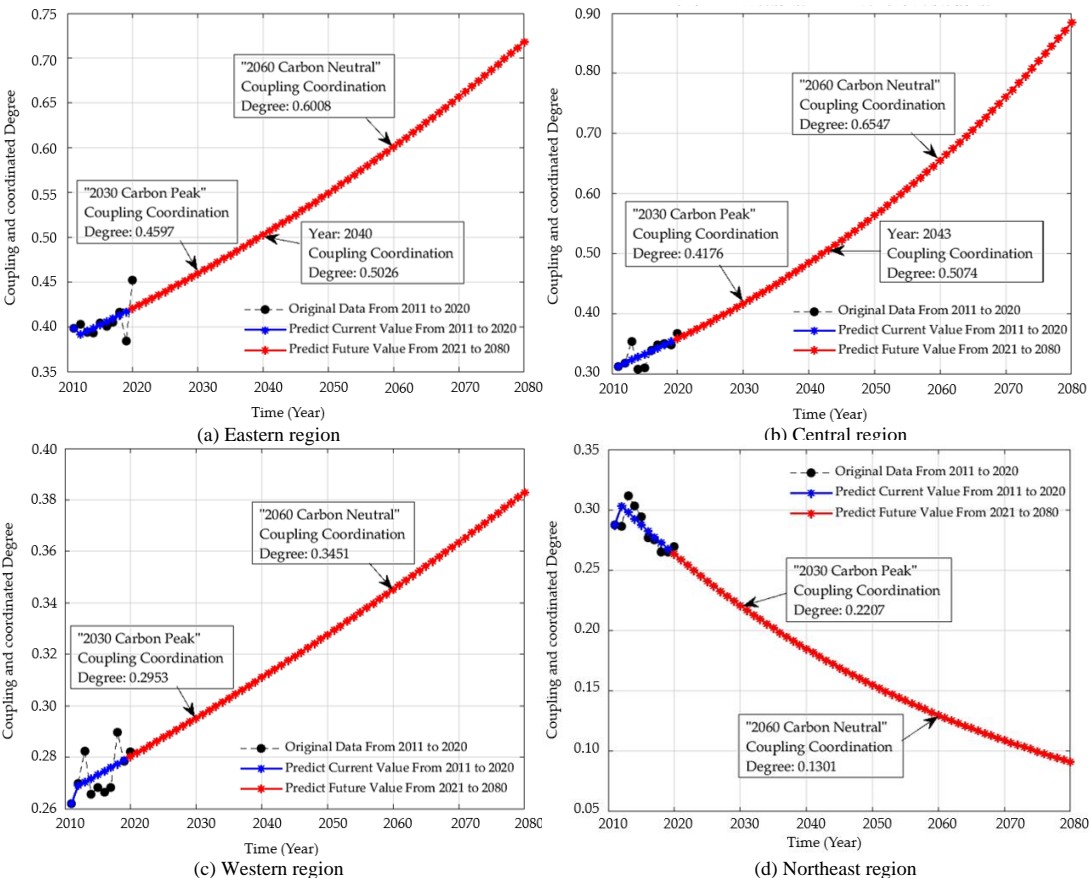

**Figure 2.** The prediction of GE system's coupling and coordinated degree from 2021 to 2080.

The results of the model operation and precision test show that the modified GM (1,1) model based on background value optimization has a good fitting effect on the predicted data.

Meanwhile, as the -a values in the above GM (1,1) model optimized based on background values are all less than 0.3, they can be used for medium- and long-term prediction. Figure 2 shows the prediction results of the Eastern region, Central region, Western region, and Northeast region.

According to the classification criteria of the coupling and coordinated degree in Table 3, the GE system starts to reach coupling coordination when the coupling and coordinated degree is 0.5. As can be seen from Figure 2a, the average coupling and coordinated degree of the GE system in the Eastern region will gradually increase in the next 60 years. However, at the current growth rate, when China reaches the "2030 carbon peak", its coupling and coordinated degree will be 0.4597, not reaching the coordination level until 2040, and it will reach 0.6008 "in 2060 carbon neutrality". As can be seen from Figure 2b, the coupling and coordinated degree of the Central region will also show a trend of gradual growth in the next 60 years. At the current growth rate, the coupling and coordinated degree will be 0.4176 at the "2030 carbon peak", which will not reach the coordination level until 2043, and it will reach 0.6547 by "2060 carbon neutrality". According to Figure 2c, the prediction result of the average coupling and coordinated degree of the GE system in the Western region shows that it will gradually increase in the next 60 years. However, because of the low starting point and slow growth, the prediction result of 2021–2080 will not reach the coordination level in the Western region, and the coupling and coordinated degree of the GE system is relatively low regarding the "2030 carbon peak" and "2060 carbon neutrality". According to the prediction results of the average coupling and coordinated degree of the GE system in the Northeast region in Figure 2d, it can be seen that, under the current development trend, the coupling and coordinated degree will

decrease instead of increasing in the next 60 years, and the coupling and coordinated degree level will continue to decrease, failing to reach the coordination level.

As the prediction results of the average coupling coordination degree of the GE system nationwide are shown in Figure 3, it can be known that, under the current strength of green finance development and environmental protection in all regions, the coupling and coordinated degree of the GE system will continue to grow at a very slow speed and reach the coordination level in 2077. This result is more than 30 years later than that of the Eastern and Central regions. Meanwhile, the average coupling and coordinated degree of the whole country does not reach the coordination level in both the "2030 carbon peak" and "2060 carbon neutrality".

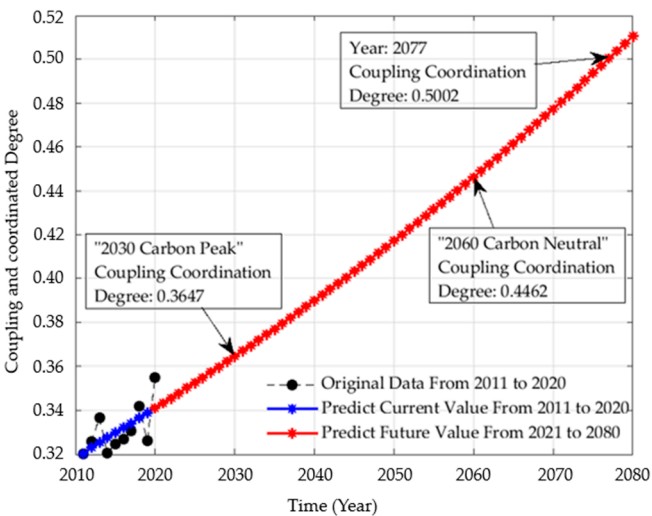

**Figure 3.** The prediction of GE system's coupling and coordinated degree nationwide from 2021 to 2080.

## 5. Conclusions and Discussion

(1) According to the calculation results of the GE system's coupling and coordinated degree, the national average level was in mild disharmony from 2011 to 2020, and, among the 30 provinces, only Guangdong and Zhejiang have reached coordination in the last ten years, while the other regions have failed to do so. China has started to develop green finance formally since 2015, but the development is still immature and there are still many problems that need to be improved [47]. However, there are significant differences in the development level of green finance among the provinces, which is closely related to the economic development degree and financial development policies of each province. However, most provinces and cities in China still pay little attention to the development of green finance [48]. Based on this, the green financial system and policies have not been highly integrated with environmental protection, resulting in the coupling disharmony of most regions' GE system and China's average GE system.

(2) Based on the prediction of the coupling and coordinated level of the GE system in the next 60 years, it is concluded that the Eastern region still continues to lead the rise and will reach the coordination level in 2040, which is the earliest among the four regions. This result is inextricably linked to the regional economic development level, environmental and climatic conditions, green finance development level, talent, and resources. The Eastern region gathers many leading economic development cities and coastal developed cities, and the overall level of economic development is relatively high, so financial development is given priority. It not only has the advantage of abundant resources but is also a gathering place for talents. Meanwhile, there are relatively advanced green finance development and mature environmental protection policies [49], so the Eastern region has the most conditions and advantages to research the integration of green finance development and environmental

protection policy. Therefore, compared with the other regions, the Eastern region has always maintained a leading level and is the first to realize GE system coupling coordination.

(3) In the absence of policy intervention, the coupling coordination level of the GE system in the Northeast region shows a gradual decline, which is the only region that does not increase but declines. It cannot achieve coupling coordination in the next 60 years and the situation is most urgent. In recent years, the overall level of economic development in the Northeast region is relatively backward, and the phenomenon of brain drain is increasingly serious [50]. At the same time, because the three northeastern provinces have once focused on the development of heavy industry, the resulting environmental pollution problem is more severe than in other regions in China. Besides, the Northeast region has not paid much attention to the development of the financial industry, and the backward financial development makes it more difficult to develop green finance. After the national plan of "Revitalizing Northeast" is put forward, the three provinces in the Northeast region gradually began industrial transformation and to formulate various green development plans [51]. However, due to the lack of human and financial resources, it is difficult for them to reverse the low and declining trend of the GE system coupling and coordinated level without policy intervention.

(4) According to the prediction results, the average coupling and coordinated level of the GE system nationwide will reach coordination in 2077, and it will not be able to achieve the coordination level within the target period of "2030 carbon peak" and "2060 carbon neutrality". However, according to the respective prediction results of the Eastern and Central regions, both regions can achieve the coordination level before the "2060 carbon neutrality" target, so the national average coupling and coordinated level must be greatly affected by the Northeast region and the Western region, which also reflects the obvious differences in development between the Northeast and Western regions and the Eastern and Central regions [52]. To improve the coupling and coordinated level nationwide, the government must first solve the two major problems in the Northeast and Western regions and implement precise and powerful policy interventions to upgrade the development of the two regions comprehensively.

(5) This paper only selects four representative indicators of the green finance system and environmental protection system, respectively, based on a large amount of relevant scientific literature, with a limited number of indicators and without considering other indicators. The GM (1,1) model based on background value optimization is selected in this paper to predict the coupling and coordinated level of the green finance–environmental protection system from 2021 to 2080 based on the consideration of accuracy and fitness, which cannot represent the prediction results of all models. In addition, this paper can only guarantee the accuracy of the predicted GE system's coupling and coordinated level in the next 60 years: the applicability of the model for predicting data beyond 60 years is not tested.

## 6. Suggestions

(1) Establish a management organization for GE system coupling and coordinated development and provide a platform for environmental information disclosure of environmental protection enterprises.

According to the measurement results in this paper, the average level of the GE system's coupling and coordinated in China has been mildly disharmonious in the past ten years. Further, according to the prediction results, coordination will not be achieved before "2060 carbon neutrality" under the current condition, which is not conducive to the process of carbon emission reduction in China. The steady improvement in green finance and environmental protection alone still reflects insufficiently on the overall coordinated development level of the GE system. It is necessary to establish coordinated development management departments through central and local governments and implement top-down management and supervision measures. The management department is specifically responsible for studying the green finance–environmental protection system, collecting data

on regional and each enterprise's green environmental protection science and technology development, pollution emission, green capital loans, and other data related to evaluation of green development and evaluating the coordinated development of the regional GE system objectively every year, and analyzing the existing problems to formulate corresponding policies and measures [53] so as to more effectively infiltrate environmental protection goals into green finance policy. At the same time, the future development direction should be planned for the actual situation of each region to narrow the gap in the coupling and coordinated level of the GE system. Moreover, the management agency needs to pay particular attention to the financing problems of environmental protection enterprises, build an authoritative and efficient information disclosure mechanism and platform, use Fintech to realize information sharing [54], and formulate unified, comprehensive, and specific information disclosure standards and mechanisms to upload and update the real and comprehensive environmental information of green enterprises timely, which is convenient for financial institutions to carry out risk and value assessment of enterprises so that these enterprises can obtain timely and sufficient financial support [55] and guide more green capital to flow into environmental protection construction, promote the integration of green finance and environmental protection effectively, and improve the overall coupling and coordinated level of the GE system.

(2) Promote industrial upgrading with green environmental protection as the orientation, and accelerate the transition from "gray economy" to "green economy".

Among the four regions of China, the coupling and coordinated levels of the GE systems in the Northeast region and Western region are relatively low. According to the prediction results, in the next 60 years, the Western region cannot reach coordination, while the Northeast region even shows a trend of decline year by year. These results are inseparable from the status quo of high-pollution and high-emission industries. For a long time, traditional industrial manufacturing has been labeled as "environmental pollution". Heavy industry is mainly concentrated in the Northeast region and Western region and spreads all over the country at the same time. However, the development of traditional heavy industry can no longer adapt to China's current demand for environmental protection construction. However, many enterprises have not realized the necessity and urgency of green transformation due to the history and experience of traditional heavy industry for many years. Today, China takes environmental protection seriously, and industrial upgrading is not only a requirement for technological progress but also a requirement for reducing pollution and emissions. First, the government needs to formulate mandatory policies for green industrial upgrading [56] to limit pollution emissions in all industries, encourage industries and enterprises to comply with the requirements, and give rewards and public praise to enterprises that significantly reduce pollution emissions while imposing fines and other penalties on enterprises that exceed the limits of pollution emissions. Second, implement top-down incentive policies; that is, the central government stimulates local governments to develop, design, and operate the industries and projects that are green, low-carbon, and sustainable, and provide financial allocation; meanwhile, local governments encourage local enterprises to make green innovations and invest emerging environmental protection projects through financial provision to promote green transformation and development of all provinces and all kinds of traditional industry. Push the transformation from the "grey economy" structure to the "green economy" structure so as to improve the coupling and coordinated level of the GE system in various regions.

(3) Build a demonstration area for the coupling and coordinated development of the GE system and give play to the radiation and driving role of high-level areas.

According to the above calculation and analysis of the GE system's coupling and coordinated level in 30 provinces, the overall level of the Eastern region is higher than that of the Western, Central, and Northeast regions. Further, Zhejiang Province and Guangdong Province in the Eastern region are the only two provinces that have reached the coordination level. Therefore, first, it is important to sum up the policy experiences of the Eastern region and set Zhejiang and Guangdong Provinces as demonstration zones for

the GE system's coupling coordination, and then research and refine the most advanced and effective practical experience of the two provinces for other provinces to learn from. Other provinces are supposed to develop scientific, reasonable, and efficient personalized solutions based on their own objective conditions development degree to ensure that local green financial development policies can better integrate with environmental protection and promote environmental protection construction so as to improve the coupling and coordinated level of the GE system. Second, the areas with low coupling and coordinated level of the GE system are often accompanied by a relatively backward green financial development, so regions can strengthen communication and cooperation with each other, and high-level areas give play to the radiation and driving role to provide assistance of manpower, resources, and policies and other assistance in many respects in order to help low-level areas develop green finance. In addition, the environmental protection goals and policies should be incorporated into the development goals and policies of local green finance so as to improve the "environmental protection inclusion rate" of green financial development and promote the coupling and coordinated level of the GE system [57].

(4) Focus on supporting the Western and Northeast regions, and improve the national average level from the weak points.

According to this paper's research and forecast results, the Eastern region and Central region's coupling and coordinated level of GE system can achieve coordination before "2060 carbon neutrality", while the national average level of coupling and coordination fails to achieve coordination before 2060, which is mainly caused by our country's two "short boards": the Western region and Northeast region. Under the current development condition, the coupling and coordinated level in the Western region increases slowly and cannot reach coordination in the next 60 years, while the Northeast region even shows a gradual decline, which directly leads to the predicted result that the national average coupling and coordinated level cannot reach the coordination level before 2060. On the one hand, the central government must focus on the economic development level, lack of resources, and brain drain in the Western region and Northeast regions and adopt special appropriation measures to provide substantial financial support to these two regions, as well as ensuring that the funds are put into practice [58]. It is important to develop local green resources and mobilize human and environmental resources from other regions to help develop the local green economy and implement environmental protection policies. On the other hand, local governments in the Western region and Northeast region need to formulate appropriate policies for the introduction of talent and environmental protection industries, including providing housing and other welfare policies for registered talents or setting up special research institutes of the green finance–environmental protection system to reduce local brain drain and cultivate and attract interdisciplinary talents proficient in finance and environmental protection. At the same time, encourage environmental protection enterprises in the Northeast region and Western region to build factories and be able to enjoy certain preferential tax policies and other policy benefits, which will not only provide employment opportunities for local people but also drive the local economy and increase the intensity of environmental protection so as to promote the industry transformation and the green financial development of the Northeast region and Western region and improve the coupling and coordinated level of the GE system.

**Author Contributions:** S.L.: Conceptualization, Data curation, Writing—Original draft preparation. G.W.: Methodology, Software. L.Y.: Visualization, Investigation, Supervision, Software, Validation. J.G. and J.Z.: Writing—Reviewing and Editing. All authors have read and agreed to the published version of the manuscript.

**Funding:** This work was supported by the Strategic Research and Consulting Project of the Chinese Academy of Engineering [grant number 2022-XBZD-09, 2022-DFZD-18], Anhui Provincial Natural Science Foundation [2208085QG225], the Open Research Grant of Joint National-Local Engineering Research Centre for Safe and Precise Coal Mining [grant number EC2021009], the Key Research Project of Natural Science in Colleges and Universities of Anhui Department of Education in 2020



[grant number KJ2020A0302], and the National Natural Science Foundation of China [grant numbers 71971003, 71704002].

**Institutional Review Board Statement:** Not applicable.

**Informed Consent Statement:** Not applicable.

**Conflicts of Interest:** The authors declare no conflict of interest.

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
