# Peer review of "Analysis and Prediction of the Coupling and Coordinated Development of Green Finance–Environmental Protection in China"

_sustainability, doi:10.3390/su14159777_

Round 1

Reviewer 1 Report

The article addresses the important dimension of environment in China but there are some additional observations to improve the quality of the paper.

It is not a good practice to use abbreviations in the abstract.

The findings of the study should be conveyed in more meaningful way in the abstract.

The introduction is without theoretical and empirical justifications. I would like to suggest to incorporate some numerical evidences highlighting the importance of the discussed issue.

The selection of sample and time period should be justified.

The discussion part should be elaborated with quoting earlier studies on the topic.

The suggestions are based on those dimensions which are not analyzed in the study like industrial upgradation etc.

Author Response

Many thanks to the reviewer for your positive recognition and full affirmation of my manuscript. We feel much honored that the reviewer made positive comments on the significance of topic selection. The comments are valuable and very helpful for revising and improving the quality of our paper. We have studied comments carefully and have made revisions thoroughly for our manuscript in accordance with reviewers' suggestions.

Point 1: It is not a good practice to use abbreviations in the abstract.

Response 1:

Thanks for your suggestions. We have deleted and corrected the abbreviations in the abstract. (see Lines 16,18,20)

Point 2: The findings of the study should be conveyed in more meaningful way in the abstract.

Response 2:

    Thanks for your suggestions. We have modified the conclusion part in more meaningful way in the abstract. (see Lines 22-25,28)

Point 3: The introduction is without theoretical and empirical justifications. I would like to suggest to incorporate some numerical evidences highlighting the importance of the discussed issue.

Response 3:

Thank you for your valuable comments. We add the data related to the present situation of Chinese coal in the introduction section, to illustrate its difficulty and importance of China to accelerate the process of environmental protection(see Lines 39-45,59-61), at the same time, we also analyze data on the development and change of China's green finance, to prove that China attaches great importance to green finance as well as the feasibility and necessity of this paper to study how to better coordinate the development of green finance and environmental protection. (see Lines 84-95)

“China's energy structure is dominated by coal, and its coal output has also steadily ranked first in cliff type in recent years [3]. According to the latest statistics released by BP World Energy Statistical Yearbook, the total global coal output in 2021 is 8.173 billion tons, while China's coal output is 4.126 billion tons, which accounts for 50.5% of the total global output, more than half. China's current energy mix also makes its environmental problems more acute, as coal is difficult to extract and use without emit-ting carbon dioxide.”

“According to data released by the National Bureau of Statistics, the share of coal consumption in China's total energy consumption has been declining year by year, and dropped by more than 10 percent in the past decade.”

“According to the data released by the Ministry of Ecology and Environment of China, as of June 2013, the balance of green credit in China is only 4.85 trillion yuan, but by the end of 2021, it has increased to 15.9 trillion yuan, ranking first in the world. Before 2015, the issuance scale of green bonds in China is almost zero, but by 2021, it has reached 440.1 billion yuan, which is the second largest green bond market in the world from the perspective of cumulative issuance and annual issuance. It can be said that the green bond market starts from zero and develops rapidly. Similarly, investment in green projects is almost zero in 2015 and before, but it has reached 5.3729 trillion yuan by 2020. It can be seen that China attaches great importance to the coordinated development of green finance and environmental protection. Therefore, although green finance starts late in China, it develops rapidly, and it is necessary, important and ur-gent to study how to make green finance better serve environmental protection.”

Point 4: The selection of sample and time period should be justified.

Response 4:

Thanks for your suggestion. The indicators in this paper are selected based on a large number of existing high-quality scientific research, and the data comes from wind database, China Environmental Statistical Yearbook, China Statistical Yearbook and other authoritative databases. The reason why we choose the data of the last decade is that China's official development of green finance starts in 2015, and after 2010, environmental protection begins to receive high and widespread attention from the world, and the goal and concept of carbon neutrality are put forward. Moreover, China has officially committed to carbon peak and carbon neutrality date in 2019. Therefore, the data related to environmental protection and green finance in the last decade are more representative and meaningful for research. Based on the above considerations, we select indicators, samples and time period. We have also added illustration in this paper. (see lines 252-258)

“China develops green finance officially in 2015, and after 2010, the concept of environmental protection and carbon neutrality began to receive more and more attention from the world, so does China, and China committees to China's carbon peak and carbon neutrality date at the United Nations General Assembly in 2019. Therefore, the data related to environmental protection and green finance in the last decade are more representative and meaningful for research. Based on the above objective reasons, this paper selects…”

Point 5: The discussion part should be elaborated with quoting earlier studies on the topic.

Response 5:

Thanks for your suggestion. We have quoted some earlier studies on the topic discussed in the discussion part to elaborate the viewpoints. (see Lines 501-504,526,542-544)

“However, there are significant differences in the development level of green finance among provinces, which is closely related to the economic development level and financial development policies of each province. However, most provinces and cities in China still pay little attention to the development of green finance [48].”

“…which also reflects the obvious differences in development between the northeast and western regions and the eastern and central regions [52].”

  1. Wang, X.H. The Study on Difference of Financial Development in China. Economic Geography 2007, 27, 183-186.
  2. Han, Z.L.; Zhang, Y.W. Temporal and Spatial Differences of the Economic Synthetic Development Ability in Northeast China. Economic Geography 2010, 30, 716-722.
  3. Li, L.; Wang, B.; Xu, J. The Analysis on the Areal Difference of Coordinating Degree between Regional Economy and Ecological Environment -- Based on Comprehensive Evaluation Method. Science and Technology Management Research 2014, 34, 38-41.

Reviewer 2 Report

While I am sympathetic to the issues investigated in the paper and do see some value in this work, I have serious reservations about it.

First, the introduction briefly overviews the paper's findings, yet it fails to provide sufficient insights into the underlying economic significance. Nor does it sufficiently discusses how these findings relate to others in the literature (see below). I would suggest that the author(s) make a bigger effort in highlighting the differences (if any) between the body of the reference literature and the current paper. It is indeed crucial to make clear at the outset, and in a non-technical manner, how the framework developed in the present paper contributes to our understanding of the expected evolution of the symbiotic relationship between green finance and environmental protection. My general advice here is intended to facilitate reading and render the paper more appealing in terms of the interpretation of the results.

Another big concern is about relevance. The original intuition upon which the present paper is built is easily found in other works, which also provide an operational (computational) content to it based on the GM(1,1) model. In particular, the paper seems very close in spirit to Wang et al. (2022), who advance a GM(1,1)-based analysis of the coupling between green finance development and carbon neutrality, and to my surprise no clear indication of how the present piece improves on the mentioned works is offered. Methodology-wise, I also believe it fails to provide a fundamental contribution relative to the mentioned work, and I struggle to see how this could be perceived to be an independent contribution to the literature. More specifically, the analysis builds upon a generalized GM(1,1) model based on background value optimization via numerical integration. While this generalization of the classical GM(1,1) model improves on prediction accuracy and estimation, it fails to model both initial value and the background values as variables to be optimized to enhance the overall fitting accuracy of the model.  As a leading example, Wang and Lu (2020) show how to how to determine the parameters of the background value and the specific form of time response minimizing the mean relative error (MRE), while at the same time optimizing the initial value on the basis of he optimized background one within a dynamic multivariate optimization setting. To what extent are the results of the present paper robust to this (or similar) model improvement? 

My overall impression is that the paper in its current form lacks sufficient advance in the state-of-the-art of the field to be accepted for publication in an outlet such as Sustainability.

Beyond these main concerns, I have a few more points, which might be useful towards building a sounder, independent contribution of the paper:
i) The author should carefully discuss how the present paper relates to Wang et al. (2022) in terms of modelling approach and findings; 

ii) There are too many instances of confusing exposition and typos, e.g. in the Abstract and Introduction. At the moment, the paper reads as an unpolished draft that is not really ready for submission to a journal. For
this paper to be widely read and to have an impact, it needs to be much more clearly written. In this respect, it would greatly benefit from deeper polishing of its exposition and proof-reading. 

 Wang, Y. and Lu, J. (2020). Improvement and application of GM(1,1) model based on multivariable dynamic optimization. Journal of Systems Engineering and Electronics, 31(3): 593-601.

Wang G, Li S, Yang L. Research on the Pathway of Green Financial System to Implement the Realization of China's Carbon Neutrality Target. Int J Environ Res Public Health. 2022 Feb 21;19(4):2451. doi: 10.3390/ijerph19042451. PMID: 35206639; PMCID: PMC8872555.

Author Response

Many thanks to the reviewer for your positive recognition and full affirmation of my manuscript. We feel much honored that the reviewer made positive comments on the significance of topic selection. The comments are valuable and very helpful for revising and improving the quality of our paper. We have studied comments carefully and have made revisions thoroughly for our manuscript in accordance with reviewers' suggestions.

Point 1: First, the introduction briefly overviews the paper's findings, yet it fails to provide sufficient insights into the underlying economic significance. Nor does it sufficiently discusses how these findings relate to others in the literature (see below). I would suggest that the author(s) make a bigger effort in highlighting the differences (if any) between the body of the reference literature and the current paper.

Response 1:

Thanks for your valuable comments. This paper is very different from the literature of Wang et al. (2022) in terms of research thrust, research method, research scope and data and so on.

First, in terms of theme, this paper mainly studies the compatibility level between green finance system and environmental protection system and aims to explore how to make the development of green finance better fit the environmental protection policies and objectives, so as to improve the coupling and coordinated level between the two systems. The study of coupling and coordinated relationship between the two systems is a main research contents in this paper, but Wang et al. (2022) only verify the positive correlation between green finance and carbon neutral and mainly aims to study how to improve the green financial system and policy to realize carbon neutral, so two articles focus on different points. Meanwhile, carbon neutral is only a part of the environmental protection system, so this paper studies more comprehensively and extensively.

Second, Methodology-wise, (1) in terms of studying the relations, this paper adopts the method of measuring the coupling and coordinated degree to analyze the complicated relationship between the two systems, and Wang et al. (2022) use correlation test to analyze the correlation between each variable of green financial and carbon reduction, so two papers are different in the methods and the requirements on result. The method adopted in this paper is more difficult, in-depth and extensive. This method’s research object is multiple systems, and the test results represent the interaction relationship and fit degree between systems, rather than the simple correlation between two single variables. (2) In terms of data prediction, this paper adopts GM (1,1) model based on background value optimization and the final prediction results are also based on this model. Wang et al. (2022) use GM (1,1) model and BP neural network model and the final prediction results were obtained by BP neural network model. Therefore, the prediction methods of the two papers are different.

Third, this paper not only studies the coupling and coordinated level between green finance and environmental protection system nationwide, but also studies the domestic 30 provinces respective coupling and coordinated level, compare and analyze measurement and prediction results between provinces, regions and in the whole country. But the study of Wang et al. (2022) is limited to the whole country. So, research scope of this paper is more specific and broader.

Forth, in terms of data, this paper chooses four indexes of the green financial system and environmental protection system respectively, carbon dioxide emissions is only one of indexes of environmental protection system selected in this paper. Wang et al. (2022) study carbon neutral only with the data of carbon dioxide emissions, and study green finance with the related data of three representative green financial products. So, the data selection in this paper is more complicated and multi-faceted. And the indicators selected in this paper are scientific and representative, which are also selected by a large number of high-quality relevant literature.

We have summarized differences between this paper and Wang et al. (2022) and emphasized this paper’s advancement (see lines 110-126):

“Wang et al. (2022) verify the driving effect of green finance on carbon neutral in China by using correlation test and use GM (1,1) and BP neural network model to predict the expected time of carbon peak and carbon neutral in China under the effect of green finance. This paper expands research object to the whole environmental protection system and selects four scientific and representative relevant indicators of green finance system and environmental protection system respectively based on a lot of scientific researches. Meanwhile, this paper subdivides the research scope to the domestic 30 provinces and divides them into four regions and they are Eastern, Central, Western and Northeast regions. Beside measuring and predicting the coupling and coordinated level of the whole country, this paper also studies, compares and analyzes the coupling and coordinated level of provinces and regions respectively. In addition, it is not to study the simple correlation between green finance and environmental protection, but studies the multiple and complex relationship between the two systems and the fit degree, which is reflected by the coupling and coordinated degree in this paper. And then the improved GM (1,1) model based on background value optimization is used to predict whether and when the coupling and coordinated level of GE system can reach the coordination level nationwide and in each region.”

Point 2: It is indeed crucial to make clear at the outset, and in a non-technical manner, how the framework developed in the present paper contributes to our understanding of the expected evolution of the symbiotic relationship between green finance and environmental protection. My general advice here is intended to facilitate reading and render the paper more appealing in terms of the interpretation of the results.

Response 2:

According to your suggestions, we have provided more insights into the underlying economic significance and reflected the expected symbiotic relationship between green finance and environmental protection in China more specifically in the introduction and combed out the research framework of this paper in more detail for readers to better understand as well see lines (100-106,126-142):

“In recent years, China has an increasingly urgent demand for environmental protection and attaches more importance to the development of green finance. However, green finance starts relatively late in China, and whether it can be well coordinated and integrated with China’s environmental protection goals and policies remains to be tested, and how to improve the promoting effect of green finance on environmental protection also remains to be studied.”

“Therefore, first of all, this paper adopts the method of measuring the "coupling and coordinated degree” and quantifies the interaction and the fit level between green finance and environmental protection in each province according to the relevant data of 30 provinces from 2011 to 2020 and calculates the average coupling and coordinated level of green finance-environmental protection system in each region and nationwide. Then, the improved GM (1,1) model based on background value optimization is used to predict the changing trend of the average coupling and coordinated level of GE system nationwide and in each region in the future and studies whether or when they will reach the coordination level. Finally, according to the research results, this paper puts forward scientific policy suggestions to improve the coupling and coordinated level between domestic green finance development and environmental protection policies. This paper aims to study how to integrate green finance development and environ-mental protection policies more effectively nationwide and in different regions, so as to make green finance better serve environmental protection objectives, improve the coupling and coordinated level of GE system, and accelerate the process of green development and the transformation from "grey economy" to "green economy" in China.”

Point 3: Another big concern is about relevance. The original intuition upon which the present paper is built is easily found in other works, which also provide an operational (computational) content to it based on the GM (1,1) model. In particular, the paper seems very close in spirit to Wang et al. (2022), who advance a GM (1,1)-based analysis of the coupling between green finance development and carbon neutrality, and to my surprise no clear indication of how the present piece improves on the mentioned works is offered. Methodology-wise, I also believe it fails to provide a fundamental contribution relative to the mentioned work, and I struggle to see how this could be perceived to be an independent contribution to the literature.

Response 3:

Thank you for your valuable comments. This paper is very different from Wang et al. (2022) in terms of research thrust, research method, research scope and data selection, and this paper is more advanced and more specific. This question has been answered above. And it’s necessary to stress that Wang et al. (2022) carry out correlation test to study the relationship between green financial and carbon neutral, not using the method of coupling and coordinated degree, and these two methods are different in the test process and the research content and so on. Correlation test is to verify the correlation between two variables, but the coupling and coordinated degree requires the selection of indicators, multi-step processing of data and relatively complex calculation to obtain the degree of interaction and the fit level between two systems. In addition, the data prediction of the two papers is completely different in methods. This paper adopts GM (1,1) model based on background value optimization for and the final prediction results are also based on this model. However, Wang et al. (2022) use GM (1,1) model and BP neural network model to predict and the final prediction results are obtained by BP neural network model. So, Methodology-wise, the two papers are completely different.

Point 4: More specifically, the analysis builds upon a generalized GM(1,1) model based on background value optimization via numerical integration. While this generalization of the classical GM(1,1) model improves on prediction accuracy and estimation, it fails to model both initial value and the background values as variables to be optimized to enhance the overall fitting accuracy of the model.  As a leading example, Wang and Lu (2020) show how to how to determine the parameters of the background value and the specific form of time response minimizing the mean relative error (MRE), while at the same time optimizing the initial value on the basis of he optimized background one within a dynamic multivariate optimization setting. To what extent are the results of the present paper robust to this (or similar) model improvement?

Response 4:

Thank you for your valuable comments. After reading a large number of literatures about GM (1,1) model’s improvement methods, this paper finally adopts numerical integration method to optimize the background values, so as to improve the accuracy of the model. This method has been recognized by a large number of scientific and high-quality literatures and widely used. Jiang et al. (2014) optimized the background value of GM (1,1) model from the perspective of integral geometric meaning by using the idea of function approximation. The results show that the GM (1,1) model established by the optimized background value calculation formula can significantly improve the prediction accuracy. Jiang et al. (2015) believe that it was an effective method to improve GM (1,1) model by optimizing the background value. Based on the core idea of Riemann integral, they optimize the background value and verify the reliability and practicability of the improved GM (1,1) model. Zu et al. (2018) use GM (1,1) model based on background value optimization to predict the future GDP data of Mudanjiang city, and compare the prediction error of GM (1,1) model based on background value optimization with that of traditional GM(1,1) model, and find that the former has better prediction accuracy than the latter. As you said, some scholars propose to improve the overall accuracy by simultaneously optimizing the initial value and background value or other methods to optimize the GM (1,1) model. Although we don’t adopt other methods like this, we have tested the accuracy of each prediction result and all the results reach the standard. Therefore, numerical integration is adopted in this paper to evaluate GM (1,1) model and the model of background value optimization is suitable for data prediction in this paper, and the predicted results are also scientific and reliable. In this paper, we add more explanation of GM (1,1) model based on background value optimization and related literature. (see lines 353-365)

“This method has been recognized by a large number of scientific and high-quality literatures and widely used. Jiang et al. (2014) optimized the background value of GM (1,1) model from the perspective of integral geometric meaning by using the idea of function approximation. The results show that the GM (1,1) model established by the optimized background value calculation formula can significantly improve the prediction accuracy [43]. Jiang et al. (2015) believe that it was an effective method to im-prove GM (1,1) model by optimizing the background value. Based on the core idea of Riemann integral, they optimize the background value and verify the reliability and practicability of the improved GM (1,1) model [44]. Zu et al. (2018) use GM (1,1) model based on background value optimization to predict the future GDP data of Mudan-jiang city, and compare the prediction error of GM(1,1) model based on background value optimization with that of traditional GM(1,1) model, and find that the former has better prediction accuracy than the latter [45].”

Point i): The author should carefully discuss how the present paper relates to Wang et al. (2022) in terms of modelling approach and findings;

Response i):

Thank you for your suggestions. We have added discussion on differences between this paper and Wang et al. (2022) and emphasized this paper’s advancement. (see lines 110-142)

“Wang et al. (2022) verify the driving effect of green finance on carbon neutral in China by using correlation test and use GM (1,1) and BP neural network model to predict the expected time of carbon peak and carbon neutral in China under the effect of green finance. This paper expands research object to the whole environmental protection system and selects four scientific and representative relevant indicators of green finance system and environmental protection system respectively based on a lot of scientific researches. Meanwhile, this paper subdivides the research scope to the domestic 30 provinces and divides them into four regions and they are Eastern, Central, Western and Northeast regions. Beside measuring and predicting the coupling and coordinated level of the whole country, this paper also studies, compares and analyzes the coupling and coordinated level of provinces and regions respectively. In addition, it is not to study the simple correlation between green finance and environmental protection, but studies the multiple and complex relationship between the two systems and the fit degree, which is reflected by the coupling and coordinated degree in this paper. And then the improved GM (1,1) model based on background value optimization is used to predict whether and when the coupling and coordinated level of GE system can reach the coordination level nationwide and in each region. Therefore, first of all, this paper adopts the method of measuring the "coupling and coordinated degree” and quantifies the interaction and the fit level between green finance and environmental protection in each province according to the relevant data of 30 provinces from 2011 to 2020 and calculates the average coupling and coordinated level of green finance - environmental protection system in each region and nationwide. Then, the improved GM (1,1) model based on background value optimization is used to predict the changing trend of the average coupling and coordinated level of GE system nationwide and in each region in the future and studies whether or when they will reach the coordination level. Finally, according to the research results, this paper puts forward scientific policy suggestions to improve the coupling and coordinated level between domestic green finance development and environmental protection policies. This paper aims to study how to integrate green finance development and environmental protection policies more effectively nationwide and in different regions, so as to make green finance better serve environmental protection objectives, improve the coupling and coordinated level of GE system, and accelerate the process of green development and the transformation from "grey economy" to "green economy" in China.”

Point ii): There are too many instances of confusing exposition and typos, e.g. in the Abstract and Introduction.

Response ii):

Thank you for your suggestions. We checked and corrected sentences and typos carefully in the abstract and introduction.

Reviewer 3 Report

Line 16: It is not usual to put the name of the software, especially not in the abstract. MATLAB needs to be deleted.

Line 287: It is necessary to delete the title of Professor Deng and it is necessary to insert the year.

Line 468: I would put section 6 before the conclusion.

Authors should emphasize in the introduction what the paper's hypotheses or research questions are.

Author Response

Many thanks to the reviewer for your positive recognition and full affirmation of my manuscript. We feel much honored that the reviewer made positive comments on the significance of topic selection. The comments are valuable and very helpful for revising and improving the quality of our paper. We have studied comments carefully and have made revisions thoroughly for our manuscript in accordance with reviewers' suggestions.

Point 1: It is not usual to put the name of the software, especially not in the abstract. MATLAB needs to be deleted.

Response 1:

Thanks for your suggestions. We have deleted MATLAB in the abstract.

Point 2: It is necessary to delete the title of Professor Deng and it is necessary to insert the year.

Response 2:

    Thanks for your valuable comments. We have deleted the title, insert the year and added it in the references part. (see line 340)

Point 3: I would put section 6 before the conclusion.

Response 3:

Thanks for your suggestions. The discussion and conclusion section of this paper is the discussion and analysis of the measurement and prediction results of the coupling coordination level of GE system, and the policy suggestions in the section 6 are targeted suggestions put forward based on the contents in the discussion and conclusion section, so it may not be appropriate to put the section 6 before the conclusion.

Point 4: Authors should emphasize in the introduction what the paper's hypotheses or research questions are.

Response 4:

Thanks for your valuable comments. We add some contents related to the paper’s hypotheses and research questions in the introduction. (see line 100-106,126-142)

“In recent years, China has an increasingly urgent demand for environmental protection and attaches more importance to the development of green finance. However, green finance starts relatively late in China, and whether it can be well coordinated and integrated with China’s environmental protection goals and policies remains to be tested, and how to improve the promoting effect of green finance on environmental protection also remains to be studied.”

“Therefore, first of all, this paper adopts the method of measuring the "coupling and coordinated degree” and quantifies the interaction and the fit level between green finance and environmental protection in each province according to the relevant data of 30 provinces from 2011 to 2020, and calculates the average coupling and coordinated level of green finance-environmental protection system in each region and nationwide. Then, the improved GM (1,1) model based on background value optimization is used to predict the changing trend of the average coupling and coordinated level of GE system nationwide and in each region in the future and studies whether or when they will reach the coordination level. Finally, according to the research results, this paper puts forward scientific policy suggestions to improve the coupling and coordinated level between domestic green finance development and environmental protection policies. This paper aims to study how to integrate green finance development and environ-mental protection policies more effectively nationwide and in different regions, so as to make green finance better serve environmental protection objectives, improve the coupling and coordinated level of GE system, and accelerate the process of green development and the transformation from "gray economy" to "green economy" in China.”

Reviewer 4 Report

The paper attempts to examine “Analysis and Prediction of the Coupling and Coordinated Development of Green Finance – Environmental Protection in 3 China”. After reviewing, I find that this paper is interesting. Green finance is more popular today, and the trends of studying this subject should be more priority.

Author Response

Many thanks to the reviewer for your positive recognition and full affirmation of my manuscript. We feel much honored that the reviewer made positive comments on the significance of topic selection. The comments are valuable and very helpful for revising and improving the quality of our paper. We have studied comments carefully and have made revisions thoroughly for our manuscript in accordance with reviewers' suggestions.

Point 1: The paper does not explain why they used GM(1,1). In this case, we can perform the panel data analysis. Why you don’t select the panel data analysis.

Response 1:

Thanks for your valuable comments.

First, we add some explanation about the reason on using GM(1,1). (see lines 339-342,344-349)

“…it predicts grey process that is time-dependent and varies within a certain range.”

“…it solves the problem of modeling continuous differential equations, which have been considered unsolvable. And grey system theory…”

“GM (1,1) model is suitable for use in a regular data change predictions, and according to the measurement results of nationwide and regional coupling and coordinated degree in this paper, they all belong to regular monotonous change. In addition, the grey prediction GM (1,1) model does not require a large number of data samples, which is suitable for the 10-year sample data selected in this paper.”

Second, panel data analysis is not selected in this paper because it only tests existing data and conducts correlation analysis and can only analyze simple correlation between various variables. This paper aims to study the relationship between green financial system and environmental protection system, not the relationship between simple variables. The "coupling and coordinated degree" can quantitatively analyze the interaction relationship and fit level between systems, that is, multiple and complex interaction relationship between green finance and environmental protection systems can be obtained through the measurement results, which is more consistent with the research purpose of this paper.

Point 2: There are many kinds of GM, but why do you select GM(1,1) but not GM(2,1) or other techniques. The paper should have more discussions about techniques.

Response 2:

Thanks for your suggestions. We select GM (1,1) model because it is more suitable for use in a regular data change prediction, and GM (2,1) model is more suitable for the nonmonotonic oscillating development sequence, or saturation S-shaped. According to the measurement results of nationwide and regional coupling and coordinated degree in this paper, they all belong to regular monotonous change. In addition, the grey prediction GM (1,1) model does not require a large number of data samples, which is suitable for the 10-year sample data selected in this paper. Therefore, GM (1,1) model is more suitable for this paper. We have added relevant explanations in this paper according to your suggestions. (see lines 344-349)

“GM (1,1) model is suitable for use in a regular data change prediction, and according to the measurement results of nationwide and regional coupling and coordinated degree in this paper, they all belong to regular monotonous change. In addition, the grey prediction GM (1,1) model does not require a large number of data samples, which is suitable for the 10-year sample data selected in this paper.”

Point 3: Which previous studies are supporting for Table 1? Some indicators that are proxied for green finance or environmental protection should be supported by previous studies.

Response 3:

Thank you for your valuable comments. Indicators related to green finance and environmental protection selected in this paper are indeed based on a large number of existing scientific studies. And in this paper, we have illustrated the sources of each indicator by listing corresponding literature. (see lines 234-251)

“In order to measure coupling and coordinated level of China's green finance - environmental protection system better, this study combines with the relevant academic research results and selects eight indexes in view of the green finance and environ-mental protection. Among them, referring to Li et al. (2021)’s research on coupling and coordinated mechanism of green technology innovation, environmental regulation and green finance, this paper selects the A - share market value of energy-saving and environmental protection enterprise and the completed investment in treatment of industrial pollution this year as the index of green financial system and the harmless treatment volume of domestic waste as the index of environmental protection system [35]. According to Sun et al. (2021)’s study on the coupling between green technologi-cal innovation and the development of green financial system, the issuing scale of green bonds and the investment of environmental health of municipal public facilities in organized towns are selected as the indicators of the green financial system in this paper [36]. Referring to Wei et al. (2021)’s study on the coupling and coordinated relationship between ecological sustainability and high quality economic development, the carbon dioxide emissions, forest coverage rate and sulfur dioxide emissions are adopt-ed as the indicators of the environmental protection system in this paper [37]. Combined with the reference of the above scholars, the indicators selected in this paper are scientific, representative and reasonable.”

Point 4: As Figure 2, it is evident that the prediction of GE system’s coupling and coordinated degree from 2021-2080.In my opinion, the prediction is too long, and it may be inaccurate. The analysis should have more discussions about this issue as well as the limitations.

Response 4:

Thanks for your suggestions. We have tested the accuracy of each prediction, and each index of each prediction is up to standard, which ensures the accuracy of the prediction results in this prediction method and shows that the prediction method in this paper is reasonable to a large extent. However, as you said, the prediction results in this paper are only predicted by using GM (1,1) model, which does have certain limitations. So we add explanation of this limitation in the discussion section. (see lines 551-554)

“Considering accuracy and fitness, GM (1,1) model based on background value optimization is selected in this paper to predict the coupling and coordinated level of green finance-environmental protection system in 2021-2080, which cannot represent the prediction results of all models.”

Point 5: If you can predict to 2080, whether this paper will continuously be exact by the year of 2180,2280? Or, how many maximum years could we predict?

Response 5:

Thanks for your valuable comments. In this paper, GM (1,1) model is used to predict the data in the next 60 years, and the accuracy of each prediction result has been verified and reached the standard. However, due to the limitations of the model itself, the accuracy may not be guaranteed by using this model to predict the data in 2180, 2280 or even longer. Therefore, relatively accurate and scientific prediction results can only be obtained during the period that can pass the accuracy test. However, this paper only guarantees the accuracy of the 60-year forecast data, and the longest time that the model can predict the coupling coordination degree is not tested in this paper, so we also illustrate the limitations in the discussion section. (see lines 554-556)

“In addition, this paper can only guarantee the accuracy of the data of the predicted GE system coupling and coordinated level in the next 60 years, and does not test whether the data predicted by the model over 60 years is applicable.”

Point 6: The limitations of this study.

Response 6:

Thanks for your suggestions. We add the limitations of this article in the discussion section. (see lines 548-556)

“This paper only selects four representative indicators of green finance system and environmental protection system respectively on the basis of a large number of relevant scientific literature. The number of indicators is limited, and other indicators are not considered. Considering accuracy and fitness, GM (1,1) model based on background value optimization is selected in this paper to predict the coupling and coordinated level of green finance-environmental protection system in 2021-2080, which cannot represent the prediction results of all models. In addition, this paper can only guarantee the accuracy of the data of the predicted GE system coupling and coordinated level in the next 60 years and does not test whether the data predicted by the model over 60 years is applicable.”

Point 7: The paper is lack of appendixes, especially results’ analysis.

Response 7:

Thanks for your valuable comments. In the conclusion and discussion section, we have analyzed the coupling and coordinated level of measurement and prediction in this paper from multiple perspectives. And according to your suggestions, we expand the analysis of the results in the discussion and conclusion part and illustrate limitations of results obtained in this paper. (see Lines 501-504,517-519, 542-544, 548-556)

“However, there are significant differences in the development level of green finance among provinces, which is closely related to the economic development level and financial development policies of each province. However, most provinces and cities in China still pay little attention to the development of green finance [48]”

“so the eastern region has the most conditions and advantages to research on the integration of green finance development and environmental protection policy.”

“which also reflects the obvious differences in development between the northeast and western regions and the eastern and central regions [52]”

“This paper only selects four representative indicators of green finance system and environmental protection system respectively on the basis of a large number of relevant scientific literature. The number of indicators is limited, and other indicators are not considered. Considering accuracy and fitness, GM (1,1) model based on background value optimization is selected in this paper to predict the coupling and coordinated level of green finance-environmental protection system in 2021-2080, which cannot represent the prediction results of all models. In addition, this paper can only guaran-tee the accuracy of the data of the predicted GE system coupling and coordinated level in the next 60 years and does not test whether the data predicted by the model over 60 years is applicable.”

Round 2

Reviewer 1 Report

N/A

Author Response

Many thanks to the reviewer for your positive recognition and full affirmation of my manuscript. We feel much honored that the reviewer made positive comments on the significance of topic selection, method adoption, sample selection, results and conclusions, which enhanced my confidence in academic research. Thanks a lot!

Reviewer 2 Report

While the paper has improved on different dimensions (presentation of results, discussion of methodology) and some obvious grammatical errors have been removed, the paper is still replete with awkward sentences and, possibly as a result of repeated rearrangement of paraghraphs, its overall structure remains somewhat confusing. I would suggest the paper undergoes one further round of editing and proof-reasing, possibly by native english speakers or some serious commercial editing service.

Author Response

Thanks for your suggestions. We have rechecked and revised the awkward every sentence in this paper and made the paragraphs and overall structure more clear.

Reviewer 4 Report

Dear Authors,

This version was strongly revised by you. I agree with most of the revisions. However, I think that you should discuss more on panel data analysis on the methodology. Second, whether with the test of 70 years, 80 years, this analysis is still acceptable? I don't think that the results based on 60 yrs is the last, maybe longer.

Thank you

Author Response

Many thanks to the reviewer for your positive recognition and full affirmation of my manuscript. We feel much honored that the reviewer made positive comments on the significance of topic selection. The comments are valuable and very helpful for revising and improving the quality of our paper. We have studied comments carefully and have made revisions thoroughly for our manuscript in accordance with reviewers' suggestions.

Point 1: However, I think that you should discuss more on panel data analysis on the methodology.

Response 1:

Thanks for your suggestions. We have analyzed panel data for each of the indicators. (see lines 261-290)

“Since China starts to develop green finance in 2015, the data collected on the is-suing scale of green bonds in 2015 and before are all zero, and in 2016, only Beijing, Jiangsu, Zhejiang and Hubei have issued green bonds, indicating that these provinces start to develop green finance earlier. Afterwards other provinces start to develop this green financial product and gradually increase the scale. The overall completed in-vestment in treatment of industrial pollution this year in all 30 provinces has been de-creasing year by year, which shows that industrial pollution in China has been reduced in the last decade and the environmental protection policy is beginning to bear fruit. The overall domestic A-share market value of energy-saving and environmental protection enterprises has been rising year by year, and the provinces with energy-saving and environmental protection enterprises have also basically maintained an upward trend, but it is worth noting that from 2011 to 2020, this index in nine provinces, including Shanxi, Liaoning, Yunnan, and Xinjiang, is 0. This is inextricably linked to the local environmental protection process. In addition, most provinces show a trend of year-on-year increase in the investment of environmental health of municipal public facilities in organized towns, but there are individual provinces that show a trend of increase followed by decrease, such as Beijing and Tianjin, which may be related to local urbanization policies. Although carbon dioxide emissions and sulfur dioxide emissions are both polluting gases, the panel data show that carbon dioxide emissions in 30 provinces are still increasing year by year, which is closely related to China's coal-based energy structure, while sulfur dioxide emissions as a whole are decreasing year by year, which also indicates that the use of sulfur dioxide generating energy has been reduced and China is trying to carry out energy transformation. The forest coverage rate and the harmless treatment volume of domestic waste in all provinces show a year-on-year decrease, and these provinces all respond positively to the national environmental protection policy and the goal of carbon emission reduction. The panel data of the eight domestic indicators selected in this paper are all characterized by changes, and the differences among the provinces can be seen through observation and analysis, and these indicators and data reflect the current situation of environmental protection and green finance development in China to a certain extent, which is indicative and representative.”

Point 2: Second, whether with the test of 70 years, 80 years, this analysis is still acceptable? I don't think that the results based on 60 yrs is the last, maybe longer.

Response 2:

Thanks for your valuable comments. As you said, the results based on 60 years is not the last and the prediction model used in this paper can predict to a much longer time. However, the main idea of this paper is based on the background of China's dual carbon strategy, which is mentioned in the introduction. And the analysis of the prediction results is mainly compared with China's "2030 carbon peak" and "2060 carbon neutral" targets. In the graphs of prediction results, the nationwide and regional levels of GE system’s coupling and coordinated at these two time points are highlighted, and the analysis is mainly based on whether each prediction can reach the coordination level before the dual carbon target. Therefore, the prediction and analysis for the next 60 years in this paper is well-founded and reasonable.

Round 3

Reviewer 4 Report

Dear Sir/Madam

I am very happy to inform to you that your revised manuscript is much improved and should be considered for publication.

Therefore, I made the acceptance.

I hope that my comments do not make you any annoyed.

Thank you.